# GeoTh: An Experimental Laboratory Set-Up for the Measurement of the Thermal Conductivity of Granular Materials

**Dimitra Rapti** [1,2,*], **Andrea Marchetti** [3], **Mirco Andreotti** [4], **Ilaria Neri** [3,4] **and Riccardo Caputo** [3]

1    Department of Chemical, Pharmaceutical and Agricultural Sciences, University of Ferrara, via L. Borsari 46, 44122 Ferrara, Italy
2    New Energies And Environment–NEA Ltd., University Spin-off, via G. Saragat 1, 44122 Ferrara, Italy
3    Department of Physics and Earth Sciences, University of Ferrara, via G. Saragat 1, 44122 Ferrara, Italy
4    Istituto Nazionale di Fisica Nucleare, Ferrara Section, via G. Saragat 1, 44122 Ferrara, Italy
*    Correspondence: cpr@unife.it; Tel.: +39-0532-974680

**Abstract:** GeoTh is a new, simple, efficient, flexible, low-cost experimental laboratory apparatus. These features make it an excellent technological tool for measuring the thermal conductivity of granular materials, e.g., soils, sand, silt, clay or artificial composites. In particular, a configuration based on the one-dimensional heat conduction in steady-state regime was designed, built, and assembled to determine the thermal conductivity of the samples. In addition, we developed two user-friendly codes; the first for acquisition of the technical data (time series of air temperature; samples temperatures; and heat power); and the second relative to the elaboration of collected data and the calculation of the physical and thermophysical parameters of each analyzed sample (porosity, bulk density, % of water saturation, thermal conductivity, thermal diffusivity). The developed apparatus allows us to temporarily carry out measurements from one to six samples either in ambient conditions or at constant temperature. The robustness of the system has been tested by analyzing and measuring numerous materials in different conditions. We also performed several tests by varying the water content (between 0% and 100% water saturation) for sandy and silty-sand samples to calibrate and test the robustness of the system as well as for verifying the repeatability of the analytical data. Finally, the obtained thermal conductivity values are compared with other dataset proposed in the literature, showing a good fit.

**Keywords:** experimental thermal conductivity set-up; steady state method; water saturation; granular material

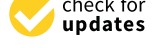



## 1. Introduction

Research on the thermo-physical parameters of materials is becoming increasingly accurate and of primary importance, with continuously increasing applications. The thermal conductivity, λ, a parameter to describe material's ability to conduct heat, is crucial for estimating the capacity, for example, of a sedimentary volume in shallow subsoil conditions to exchange and/or store the thermal energy. Indeed, quantifying this capacity is of the utmost importance in energy investigations, environmental and soil science, agronomy, geoengineering, and in the study of insulating and non-insulating materials.

In agronomic practice, for example, the seed germination and crop growth are strongly affected by their surrounding climate conditions at the seedling, which is influenced by soil thermal properties [1]. In particular, soil thermal properties (temperature, thermal conductivity, thermal diffusivity) and water content are dominant drivers to (i) soil respiration, (ii) greenhouse gases fluxes ($CO_2$, $N_2O$ emissions and $CH_4$ uptake) as far as the optimal temperature and moisture contribute to increase the decomposition of soil organic matter, (iii) the nitrification and denitrification of the soil, and (iv) the metabolism

of methane-oxidizing bacteria. Moreover, soil thermal conductivity plays a role in several remote-sensing based approaches for estimating soil moisture across large regions and it is an important physical parameter in modelling land surface processes and for land surface modelling [2–9].

In addition, when planning shallow (low enthalpy) geothermal systems, thermal response tests (TRT) are indeed commonly carried out to estimate the effective thermal conductivity of the underground. While this is a largely applied method, it has some major shortcomings, namely the fact that the test provides a unique value representing the average thermal conductivity of all lithologies crossed by the drilling and hence by the installed probe. As a consequence, in the case of large geothermal plants, these estimates could lead to underestimating the real conditions, therefore making the planned geothermal system less efficient. On the other hand, in case of overestimation, the drilling will be likely deeper than necessary, therefore adding useless investment costs. Other potential problems of this method are due to the fact that the instrumentation must be exposed outdoors, and the test needs to be performed for many hours to assure a sufficient reliability, commonly for at least 72 h. Another possible source of error in this procedure could be caused by the heat loss (or gain) induced by the different ambient temperatures, both daily and seasonally. In contrast, in laboratory conditions, there is the possibility of more accurately investigating the thermal conductivity of each material sampled for coring. The scale is different, smaller, but the accuracy is much higher, and the amount of material needed to carry out the tests is very similar to that collected during coring.

In order to contribute to this topic, we designed and realized a laboratory for measuring the thermal conductivity of unconsolidated natural materials (but possibly also synthetic ones) with primary porosity; in principle, any granular material. In addition, it is possible to recreate boundary conditions similar to real in situ conditions, such as the degree of saturation and the lithostatic pressure. In the present note, we describe the technical characteristics of the apparatus and, following several tests, the reliability of the results and the flexibility of the instrumentation. For this purpose, we also investigated the role played by the grain size distribution and the amount of water content in the tested materials whose importance has already been emphasized [10–15]. The influence of the mineral composition, the bulk density, and the porosity constitute the focus of an ongoing research project and the results will be published in a future paper.

*1.1. Heat Transmission Principles*

The thermal conductivity property of a material is defined as the amount of heat transferred, due to a unit temperature gradient, in the unit time under steady-state conditions in a direction normal to a surface, when heat transfer is dependent only on the temperature gradient [16]. This thermo-physical parameter could be determined using either steady-state methods, which measure thermal properties by establishing a temperature difference that does not change with time, or transient techniques, which usually measure the time-dependent energy dissipation process of a sample. Each of these techniques has its own advantages and limitations and it is suitable for only a limited range of materials, depending on the thermal properties, sample configuration, and measurement temperature [8].

The thermal conductivity of granular and porous materials has been extensively investigated since decades (e.g., [10,17–23] among many others).

The overall thermal behavior of a rock or sediment volume depends on several factors such as temperature, pressure, mineral content, grain-size distribution, texture, porosity, composition and properties of the fluids and gases that fill the pores. Values of thermal conductivity vary widely from rocks, sediments, and pore-filling materials.

Several empirical relationships between some of these parameters have been proposed in order to predict the thermal conductivity for different material conditions and parameters combinations (e.g., [11,24–26]). More recent analytical solutions capture salient effects on thermal conductivity, such as effective stress and Hertzian contact [27], interstitial water

and decreased contact resistance [28], and partially saturated spherical bundles with water meniscus at the contacts. Moreover, the primary processes of particle-level heat transport in granular materials, according to the processes identified by [29], show that the heat transfer explains the ordered sequence of typical thermal conductivity values:

$$\lambda_{\text{air}} < \lambda_{\text{dry-soil}} < \lambda_{\text{water}} < \lambda_{\text{saturated-soil}} < \lambda_{\text{mineral}}.$$

Notwithstanding the different technical solutions (see next section), the methodological approaches based on steady-state conditions stand on two basic models: the *infinite line source* (ILS; [30]) and the *guarded hot plate* (GHP). For example, the direct estimation of the thermal conductivity via the infinite line source model can be achieved by plotting the experimental measures of the mean temperature of the heat carrier fluid circulating within a geothermal heat exchanger versus the logarithm of time. Indeed, following a first transient phase, steady-state conditions can be assumed to occur when the semi-log relationship between heat carrier fluid temperature and time shows a linear trend and therefore analytical solutions could be applied.

When using a guarded hot plate method, a different analytical approach is applied. Following the Fourier's law, local heat flux density, $q$ [W/m$^2$], is linearly proportional to the thermal conductivity, $\lambda$ [W/m·K], and the temperature gradient, $\nabla$ T [K/m].

$$q = -\lambda \cdot \nabla \text{T}, \tag{1}$$

In other terms, thermal conductivity could be represented as a function of the input power ($Q$), the area crossed by the heat flux ($A$), and the temperature at two points ($T_1$ and $T_2$ and their difference, $\Delta T$) aligned along the flux direction with distance $\Delta L$

$$\lambda = (Q \cdot \triangle L)/(A \cdot \triangle T). \tag{2}$$

### 1.2. Other Measurement Approaches

There are many papers in the literature dealing with the calculation of thermal conductivity using different methodologies. For example, for real case in situ measurements of geothermal systems consisting of borehole heat exchangers from several tens to some hundreds of meters long, a thermal (or ground) response test (TRT or GRT) is commonly applied. The principle at the base of a TRT is the line source (or cylinder) model, which adopts the analytical solution of the heat transfer problem between the probes inside the borehole and the neighboring subsoil volume, assuming it to be laterally infinite. The approach in the field requires some simplifying assumptions on the geometry of the borehole and the pipes and allows to obtain an effective thermal conductivity value of the entire investigated rocks/sediments volume, which obviously represents a weighted average of the whole drilled stratigraphic succession. Accordingly, any information about the real thermal conductivity values of each encountered lithology and its weight (viz. thickness) in terms of contribution to the estimate of the thermo-physical parameter is lacking.

In order to obtain information on the thermal conductivity relative to each lithological contribution, however, laboratory tests are commonly performed. For this purpose, a variety of apparatuses have been proposed allowing to analyze different materials, from loose deposits to hard rocks, and with different conditions (e.g., dry/wet, or with/without confining pressure, etc.).

A crucial issue in laboratory investigations is represented by the dimensions of the tested volumes which could largely vary in the literature along with the resolution and the degree of uncertainty of the obtained results with the different set-ups.

For example, [31] used a tank with dimensions of 9, 6, and 4.5 m (length, width, and height, respectively) to reproduce a large-scale stratification and an aquifer. The set-up also includes 4.3 m-long probes with a diameter of 10 cm, therefore with a length-to-width ratio of 21.5. The tank is filled with sand of three different grain sizes, i.e., fine, medium, and coarse, and a groundwater flow is also introduced in the system. The pitfalls of this

apparatus are basically the same of the in-situ tests, namely (i) the resulting value of the thermal conductivity represents an average of all crossed sedimentary 'units', (ii) the testing apparatus is similar for that used for field operations (and thus relatively expensive), and (iii) experiments are long-lasting considering the set-up procedure and the heating and cooling phases.

In addition, [32] used a very large laboratory consisting of a wood made 18 m-long sandbox with a square section of 1.8 m by side. The box is waterproof and covered on the top. The testing technique is similar to the previous one (hydraulic pump and heater) and uses a single-U pipe as a heat exchanger. An 18-m-long aluminum tube with 12.6 cm of inner diameter is then inserted horizontally at the center of the box simulating the perforation wall, while inside is positioned the heat exchanger. Further, in this case, the thermal response test requires tens of hours. Compared to the set-up of [31], the one proposed by [32] allows one to determine either the thermal conductivity of the soil and the thermal resistance of the borehole (including the grout).

Although smaller with respect to the previous ones (ca. 1.8 × 1.8 × 2.1 m), the laboratory created by [33] is also devoted to testing quasi real scale conditions. Unlike the previous examples, however, the apparatus was set-up for testing pile-anchored geothermal systems within a sandy body. Accordingly, a precast concrete pile (1.4 m-long and 0.1 m in diameter) was inserted for ca. 1.2 m within the sand. The concrete pile contains a single-U PVC probe for the circulation of the heat carrier fluid, which was kept at a constant temperature at the entry point. It should be noted that the thermal conductivity values of the infilling sand obtained using different calculation methods show considerable differences, while a major limitation is the length of the tests (up to 7 days).

A slightly smaller (1.0 × 0.4 × 0.4 m) laboratory apparatus was set up by [34] for obtaining the thermal conductivity. However, the major difference with respect to the previously described ones is represented by the hybrid approach they followed, partly based on analogue experiments and partly by numerical modelling. Indeed, the thermal box was filled with porous material (gravel and sand) and tested with different heating linear sources. Tests were also carried out at different degrees of saturation and either with and without water flow crossing the sample. The target of these preliminary analogue tests was to investigate and reconstruct the induced heat fluxes within the samples by means of monitoring temperature and humidity at selected positions and distances from the sources. Based on the measured values, it was thus possible to reconstruct a 3D physico-thermal model of the samples that was considered as a reference for the back analysis, where numerical modelling was used for reproducing the experimental results, therefore inferring the thermal conductivity, among other parameters.

A new commercial device, though modified by the authors, has been used by [35] for testing granular material with different wet conditions. The apparatus, a guarded hot plate system, is based on a steady-state method. It works by imposing and maintaining a constant temperature difference (commonly 10 °C) between three parallel plates (a cold one in the middle and the hot ones outside), thus generating a symmetric heat flux across the two interposed samples. Considering the thickness of the tested samples, after 6–7 h, the system reaches a thermal equilibrium and the continuously measured thermal conductivity stabilizes.

Among the small-scale laboratories is the one set-up described by [36] consisting of an acrylic cylinder with an inner diameter of 0.3 m, containing a 0.2 m-thick silica sand layer with an average grain size of 0.2 mm. Water on both sides of the sandy layer separated by perforated acrylic panels and nonwoven fabrics, together with a regulated pump, allow to reproduce a constant groundwater flow. The apparatus is then completed by a thermal probe consisting of a steel electrical resistance (buried in the middle of the sandy layer perpendicular to the water flow direction) and several Pt-100 sensors. Accordingly, based on this apparatus, the authors could investigate the influence of water velocity on the thermal response of the system.

Alternative approaches for testing the thermal conductivity of natural materials make use of a commercial needle probe assembled system with a thermocouple and heating wire inside. Similar to the TRT, but at a much smaller scale, the approach is based on the thermal transient phenomenon. Such instruments are commonly portable and allow obtaining a measurement in a few minutes, however at the expense of data accuracy. Moreover, each measurement is representative of a very small volume. In the literature, there are several applications of this method devoted to analyze the influence of different variables, e.g., grain size, mineral content, instantaneous porosity, water content, density, vertical effective stress, and packing effects (e.g., [12,13,22,37–39]). For this purpose, small containers, up to few tens of cm per side, are commonly built, but direct field measurements could also be carried out in open-air conditions.

## 2. Materials and Methods

### 2.1. Custom-Built Experimental Laboratory Set-Up

For the purpose of investigating the thermal conductivity of loose materials, mainly sediments, of diverse characteristics and in different physico-chemical conditions, we designed and developed a specific laboratory apparatus trying to respond to some major constraints: (a) the use low-cost instrumentations and parts, (b) the flexibility for different settings, and (c) obtaining as accurate results as possible.

The overall scheme of the apparatus is represented in Figure 1. The system is composed of some basic parts among which are the casing for the samples thermally isolated and equipped with several sensors, a heating system with adjustable power supply, and a governing hardware and software system, including the datalogger and the general management.

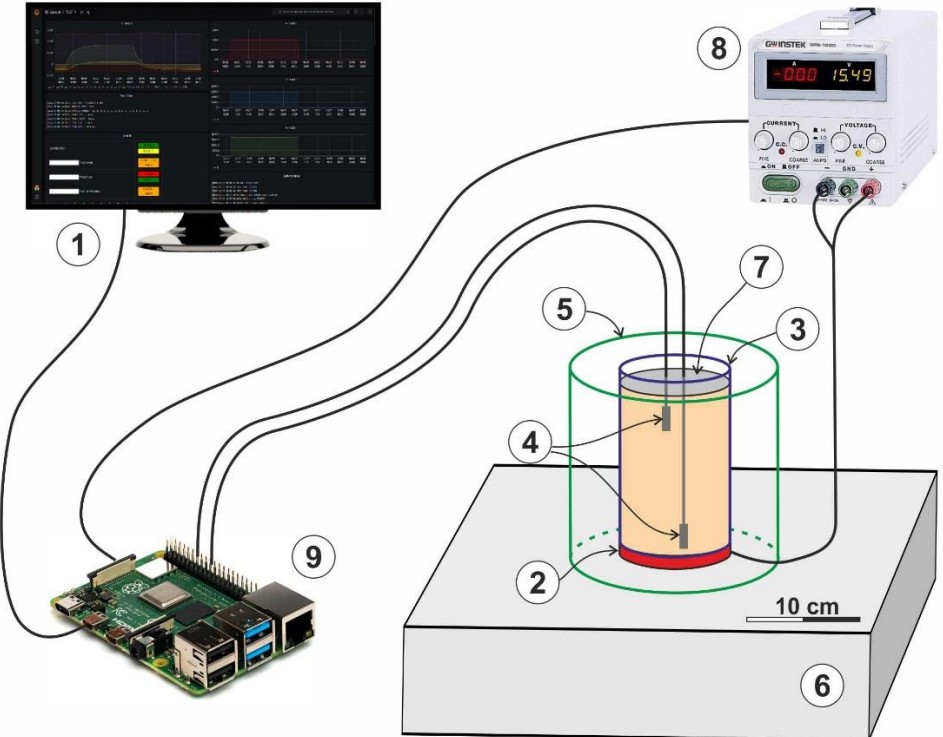

**Figure 1.** Schematic representation of the apparatus described in the present paper. (1) Graphical user interface; (2) disk-shape heat source; (3) sample casing; (4) temperature probes (PT100); (5) lateral polyurethane foam ring; (6) bottom polystyrene layer; (7) aluminum film; (8) power supply; (9) single board computer (Raspberry PI) for data acquisition, processing and storage.

### 2.2. Sample Casing

In a preliminary experimental phase, the characteristics of the sample casing were calibrated for testing an amount of material comparable to a portion of a core, say 10–20 cm-long and ca. 10 cm in diameter. In order to minimize edge effects, cylindrical sample holders were selected consisting of polyvinyl-chloride (PVC) ($\lambda = 0.159$ W/m·K). The cylinders (n. 3 in Figure 1) have internal and external diameters of 8.6 and 9.0 cm, respectively, and in this preliminary experimental phase they are 10 cm-long. The cylinders are close at the base with a 3 mm-thick aluminum disk ($\lambda = 200$ W/m·K) sealed with a high temperature resistant conductive paste in order to make the casing capable of containing water.

During the tests, each casing is posed in direct contact with a discoidal electrical resistor perfectly matching the diameter of the cylinder (n. 2 in Figure 1). The whole set up is posed on top of a 10 cm-thick layer of polystyrene ($\lambda = 0.034$ W/m·K), which acts as an insulating material (n. 6 in Figure 1). In addition, all around and in direct contact with the PVC cylinder, we fixed a polyurethane foam ($\lambda = 0.022$ W/m·K) with the same insulating purpose (n. 5 in Figure 1).

Once filled the casing with the sample material to be tested, an aluminum film is commonly posed on the top for impeding (or at least minimizing) the water evaporation during the wet tests as a consequence of the heating let into the sample and released by the electrical resistor. On the other hand, the very high conductivity of the film and its minimal thickness (i.e., negligible thermal resistance) do not alter the heat flux from the sample to the air and hence the results of the tests.

### 2.3. Hardware Components

The hardware components of the system consist of a master device single board computer, a power supply for the heater with two independent channels, and a custom-made electronic motherboard which accommodates sixteen commercial daughter-cards for temperature measurements.

Communication between the master device and power supply is based on standard SCPI protocol via USB interface, while the temperature measurements are acquired from the daughter-cards using SPI communication with multiplexed technique implemented on the motherboard. A representative scheme of hardware components and their connections is shown in Figure 2.

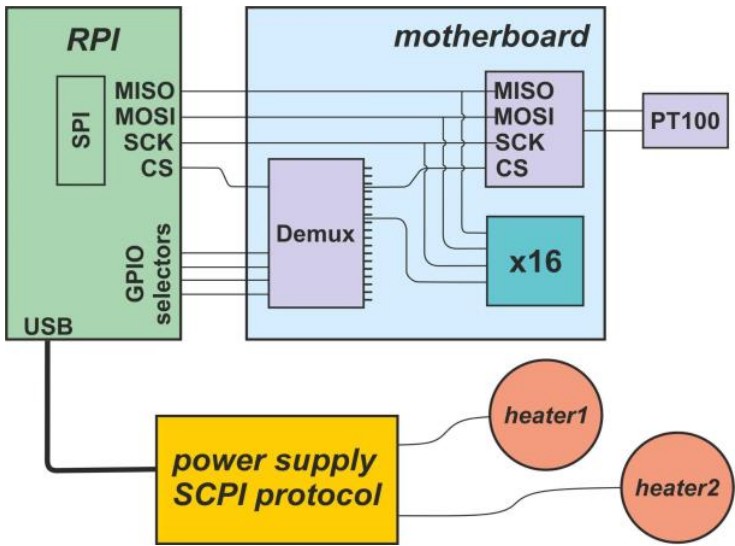

**Figure 2.** Schematic overview of the hardware components.

For governing the whole apparatus, a commercial Raspberry PI single board computer (n. 8 in Figure 1) has been exploited as a master system, because the hardware peripherals equipment, e.g., USB and SPI interfaces, is ready to use for this scope. Moreover, the Linux operating system running on the Raspberry PI allows the highly reliable implementation of custom software, it has high flexibility for low level diagnosis and configuration, various free license extensions, and a lot of information is available from the Raspberry community that could contribute to the development of custom applications.

Additionally, low-cost hardware components, such as the ones we used, allow us to have a backup system or an entire replica with a very low economic impact. For this project, a Raspberry PI 4 model B with 4 GB RAM and 32 GB SD card and official raspbian OS has been selected. The OS is installed on the SD card, but in order to prevent the possible loss of integrity, as a good practice for long term applications, two USB-disks have been configured for data storage. One linux file system formatted disk is dedicated to the database local storage, while the second one Ex-FAT file system is used as backup of the database and for pre-processed data. In the following they will be referred to as DB-DISK and BACKUP-DISK, respectively. The linux file system format for DB-DISK was set up for easy management of permissions and for security reasons, while the Ex-FAT format for BACKUP-DISK has been preferred for allowing users to have an alternatively easy way to retrieve data on different operating systems PCs.

In order to reach as constant as possible heating power during the first phase of tests and with the aim to have as low as possible economic impact factor on the system, a low-cost commercial power supply has been identified in the DC programmable Power Supply model RSPD3303X provided by RS Components SRL. This model comes out in two versions: 10 mV/10 mA and 1 mV/1 mA voltage and current precisions, respectively. Typical heating power needed for the tests is of the order of few watts. A convenient way to heat samples is the use of commercial 75 Ohm heating resistors. From these considerations, the second version of Power Supply allows to obtain a power precision of about 15 mW, while the first one shows one order of magnitude higher precision, which is not necessary for this aim.

For precision temperature sensing, Platinum Resistance Temperature Detectors (RTDs), with a resistance of 100 Ohm at 0 °C, have been chosen as temperature sensors, i.e., the so-called PT100 (n. 4 in Figure 1). These have been used for many years to measure temperature in laboratory and industrial processes, and have developed a reputation for accuracy, repeatability, and stability in time. Because these elements are usually quite fragile and our test conditions could be hard, the acquired PT100 probes are sheathed and protected by a thin film which assures a short response time due to the bare measuring resistance and a reduced thermal capacity.

In order to get precision and accuracy out of the PT100 RTD, the Adafruit MAX31865 module has been used, this being very suitable for the precise measuring of extremely high and low temperatures. Indeed, the MAX31865 IC is an easy-to-use, resistance-to-digital converter optimized for platinum RTDs. It features an amplifier which is designed to read the low resistance and can automatically adjust and compensate for the resistance of the connecting wires, together with a 15-bit ADC converter providing a nominal resolution of $\pm 0.03125$ °C. In Figure 3, results from a test performed on a PT100 probe kept at constant temperature for about half an hour are shown. Temperature as a function of time and its distribution show a resolution of 0.03382 °C, which is comparable to the nominal value within nonlinearity effects.

An external resistor sets the sensitivity for the RTD being used and a precision delta-sigma ADC converts the ratio of the RTD resistance to the reference resistance into digital form. In this project, a Rref of 430 Ohm was chosen.

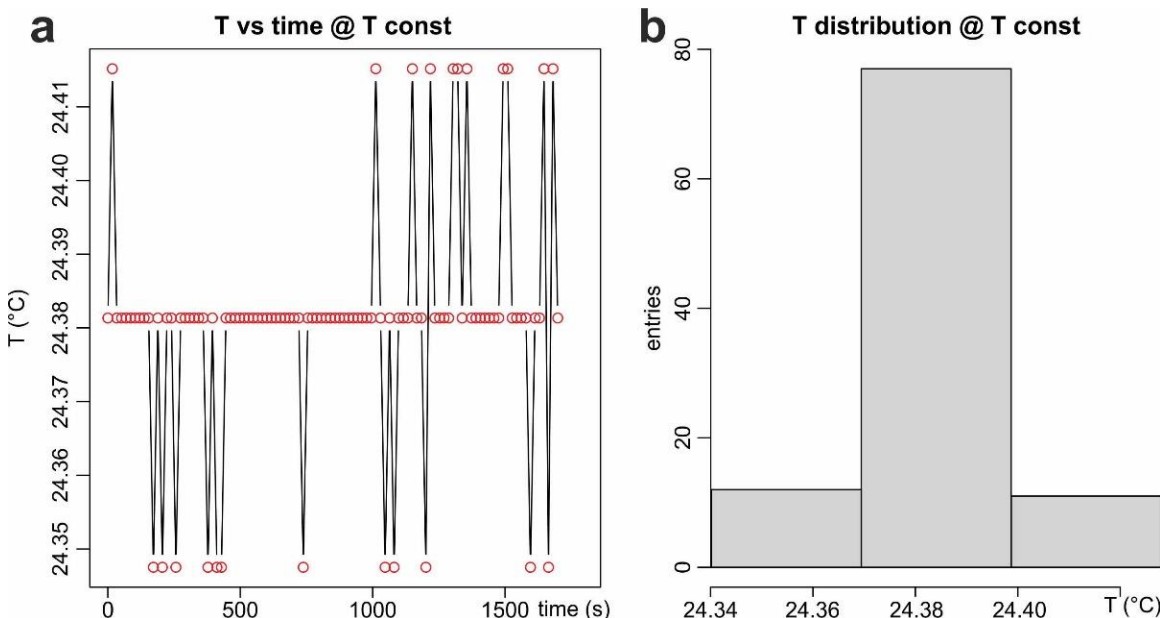

**Figure 3.** (**a**) Constant temperature versus time. (**b**) Measured temperature values distribution.

Since temperature probes could show different values due to intrinsic fabrication and due to effective Rref values, a calibration was carried out to correct temperature measurements for each sensor. As reference temperature, the mean value of all sixteen probes was assumed, while all single measurements were corrected to fit the same values. The calibration is intended to be applied offline during the data analysis phase. For the actual set of probes, differences between linear coefficients are of the order of $10^{-3}$, comparable with the difference found between nominal and measured resolution, as mentioned above. In our case, differences between the temperature offsets are of the order of $10^{-1}$ °C, but this could be very different depending on probe type. In Figure 4, temperature measurements as a function of reference temperature were reported before and after correction.

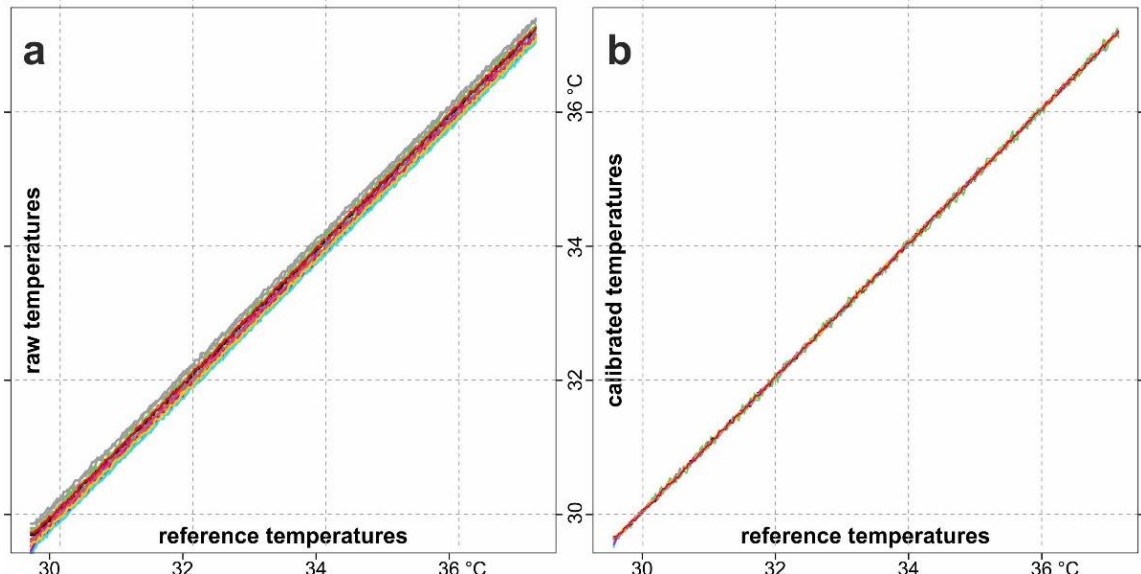

**Figure 4.** (**a**) Raw temperatures versus reference temperatures. (**b**) Corrected calibrated temperatures versus reference temperatures.

There are four screw terminals on the Adafruit MAX31865 module, so different PT100 probe types can be used with this design. Moreover, it can work with 2-, 3-, or 4-wire PT100 probe types. Since the 4-wire probe connection eliminates errors due to cable resistance, for very precise readings of low resistance values, 4-wire measurement were carried out.

With the aim to realize the whole temperature system, a printed circuit board was designed and produced at the INFN Electronics Design Service at Physics and Earth Science Department (Ferrara, Italy). The motherboard hosts the Raspberry PI and 16 Adafruit MAX31865 resistance-to-digital converter modules which communicate through SPI interface with the main controller. In order to reduce the number of chip select signals on board of the Raspberry PI, a 1-to-16 demultiplexer (74HC154) was mounted on the electronic motherboard. This allows the user to select up to 16 modules, meaning that 'contemporaneous' measurements of up to 16 temperature values can be achieved.

### 2.4. Software and User Interface

The control and monitoring system is based on a main master script, in the following called Master Script, which interfaces with hardware (power supply and motherboard; nn. 7 and 8 in Figure 1) and with a dedicated database for an organized data storage. In the same database, the single board computer monitor status is stored by a secondary script called RPImon Script. Master Script and RPImon Script are based on python3 (https://www.python.org, accessed on 10 November 2022) and run on the Raspberry as two independent daemons. In order to avoid the loss of data, two linux cron jobs have been configured to provide local and remote time scheduled data backup. Data and test configuration as well as the monitoring status are saved on the time series Influx database (https://www.influxdata.com, accessed on 10 November 2022), which is organized in six main sections, as it will be described later. User interface and data visualization are realized with Grafana (https://grafana.com, accessed on 10 November 2022) tools which interface with the Influx-DB. In Figure 5, a schematic overview of the script and used software tools is shown.

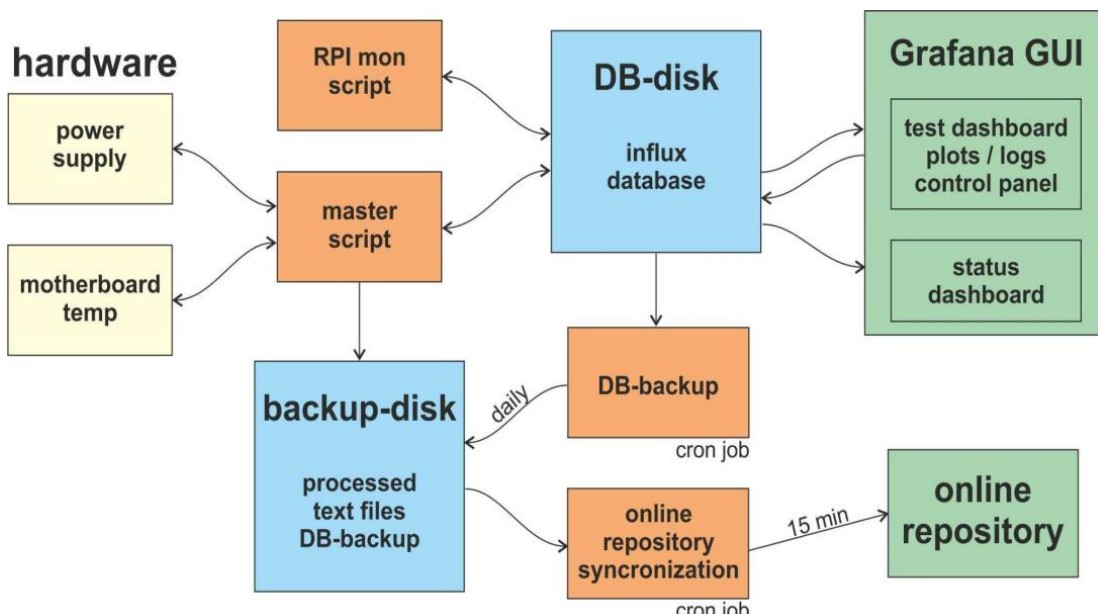

**Figure 5.** Schematic overview of software components, data fluxes and hardware interfaces.

A detailed description of each software component is reported in the following. In order to help readers to understand data and command fluxes, we start describing the database configuration. In order to have a complete view of data and system status, all information has been organized in a time series influx database configured with the following dedicated sections.

1.  *Raspberry PI System Monitor* section contains diagnostics of the single board computer, e.g., CPU temperature and usage, RAM usage and disk usage. This information allows users to investigate possible system failure.
2.  *Daq Log database* section contains all runtime information about operations executed by Master Script. This information allows users to check test configuration, correctness of hardware and database accesses, and view history of test operations.
3.  *Temperature* section contains all sixteen temperatures measured with a default time sampling interval of about twenty seconds.
4.  *Tests Configuration* section contains all details of test configuration, e.g., test name, enabling test, enabled temperature probes, heating power, enabling power, single measurements time sampling, and time interval for recording temperature average for raw data file.
5.  *Tests Data* section contains related enabled temperatures and power measurements details.
6.  *Run State* section contains information on changes of test configuration and Raspberry shutdown request. Changes on test configuration by user interface are recorded on a single flag in this section and this will trigger Master Script to reconfigure the test with the new set-up.

For data storage and management, all sections have read/write access from Main Script and RPImon Script. For data visualization, Sections 1, 2, 3, and 5 have only read access from Grafana user interface. Meanwhile, in order to allow users to control tests, Sections 4 and 6 have write access from a dedicated panel on Grafana user interface.

Main Script is designed to have full control of involved hardware and to manage data and test configuration. The script is based on five concurrent processes, each dedicated to a specific task, while a main independent process checks the running status of all tasks. Main Script processes can be summarized with the following list:

*   *Main controller process* checks running status of all other five processes, sends infos to Daq Log database section and provides a restart of Main Script in case of failure;
*   *Daq process* is dedicated to temperature measurements and control of the heating power system through the power supply. If at least one of the two possible tests is enabled, all sixteen temperatures and the two channels power supply parameters are acquired at a defined sampling rate and stored in the Temperature section of the database; temperatures are acquired by daughter-cards with SPI protocol with a multiplexed technique driven by GPIO of Raspberry PI; monitor and control of power supply are provided with standard SCPI protocol via USB interface.
*   *DataProcessing process* provides a data remap to related tests and fills the corresponding Tests Data database section.
*   *DataMean process* provides average measurements on a user defined time interval and produces raw data text files.
*   *CheckConf process* checks changes in configuration requested by users, then provides a reconfiguration of tests when required.
*   *CheckPowerSupply process* checks the correctness of connection with the power supply for the heating system.

RPImon Script is a simple single threaded process which is demanded to log into database infos about Raspberry system status. In detail, system functionality is monitored by looking at CPU temperature and usage, RAM, DB-DISK and BACKUP-DISK usage. Infos are checked and stored every minute.

Local storage into database performed by Master Script and RPImon Script is located on the external USB disk (DB-DISK). A secondary external USB disk, named BACKUP-DISK, contains processed data and daily updated database backup produced by a dedicated linux cron job running influxdb backup commands. In order to enforce integrity and safety of data, a second linux cron job running every fifteen minutes takes care of the synchronisation between BACKUP-DISK and an online data repository.

A locally installed, open-source version of the Grafana framework has been used to create the GUI (graphical user interface) for the system. GUI is composed of three

dashboards, two dedicated for independent tests and one used for the status overview visualisation. GUI dashboards are connected to the influx database used for data archive. Test dashboards show plots of enabled temperatures, heating power, applied voltage and heating resistance values versus time (Figure S1 in Supplementary Material). Time interval visualisation can be changed by users. A text panel shows the actual test configuration and status. A second text panel shows log messages from Master Script (System Status in Figure S1).

In addition, an html panel based on a javascript is dedicated to control the system. From this control panel, it is possible to configure the test and to control its execution. Javascript is used to send configurations to the Test Configuration and Run Status database sections.

The control and monitoring temperature system is designed to calibrate the power directed to the heater(s) and to measure and store the temperatures from the sensors. The system is capable of operating up to two independent sets of tests by driving two independent sets of heaters and to monitor up to sixteen temperatures. The system could be completely controlled remotely, and the online monitor allows to check the status of tests and diagnosis of the whole system in real time.

## 3. Tests for the Reliability of the Results

Following several tests for checking all components of the apparatus and validation of the whole experimental set-up, including a smooth laboratory procedure, we performed numerous further tests for verifying the reliability of the measurements and of the overall results. In particular, we verified the possible effect of the ambient temperature on the measurements, the need and the positive impact of properly insulating the sample casing, the reproducibility of the obtained measurements, and the calculated thermal conductivity values, and finally compared a set of our preliminary results with literature data.

### 3.1. Effect of Ambient Temperature

As mentioned above, the whole apparatus was purposely created to test different loose granular materials in various boundary conditions. One of these conditions is represented by the ambient temperature, which is continuously measured during the experiments by two distinct sensors. This issue was analyzed to recognize a possible impact on the final thermal conductivity values. Accordingly, several tests were performed under nearly identical thermo-physical conditions (grain size in the range 0.5 to 0.25 mm, porosity in the range 40–44%, and a bulk density of 1450 kg/m$^3$), but at two different ambient temperatures of +25 °C and −5 °C. For the former conditions, the daily temperature change in the laboratory is typically ±0.5 °C, while within the insulated chamber of the refrigerator temperature variability is limited to ±0.25 °C.

Within each tested sample, two sensors are installed at 7 cm and positioned in correspondence with the cylinder axis, with one sensor at distance of 1.5 cm from the casing bottom and the other sensor in the upper part of the sample, but remaining buried 1.5 cm from the top surface. After a few hours following their placement within the sample and before heating, the installed sensors reach a thermal equilibrium showing a temperature difference of less than 0.07 °C, thus confirming the lack of heat flux inside the insulated sample casing in both ambient temperature settings.

### 3.2. Lateral Insulation Tests

In order to verify the need and the efficiency of the lateral thermal insulation and hence the occurrence of a unidirectional (upwards) heat flux across the sample, we set-up some tests with and without a thermally resistant coating. For this purpose, we measured the temperature variation during time at two different sensors located respectively near the top ($T_A$) and the bottom ($T_B$) of the sample casing. Experiments were carried out within the refrigerator because of the more stable ambient temperature leaving the samples after a heating test to homogenize with the ambient temperature of circa −4 °C. The

results from one of such double tests are shown in Figure 6, representing the curves of the temperature versus time for the two sensors, which are characterized by initial cooling phase temperatures of ca. 17 and 2 °C, respectively.

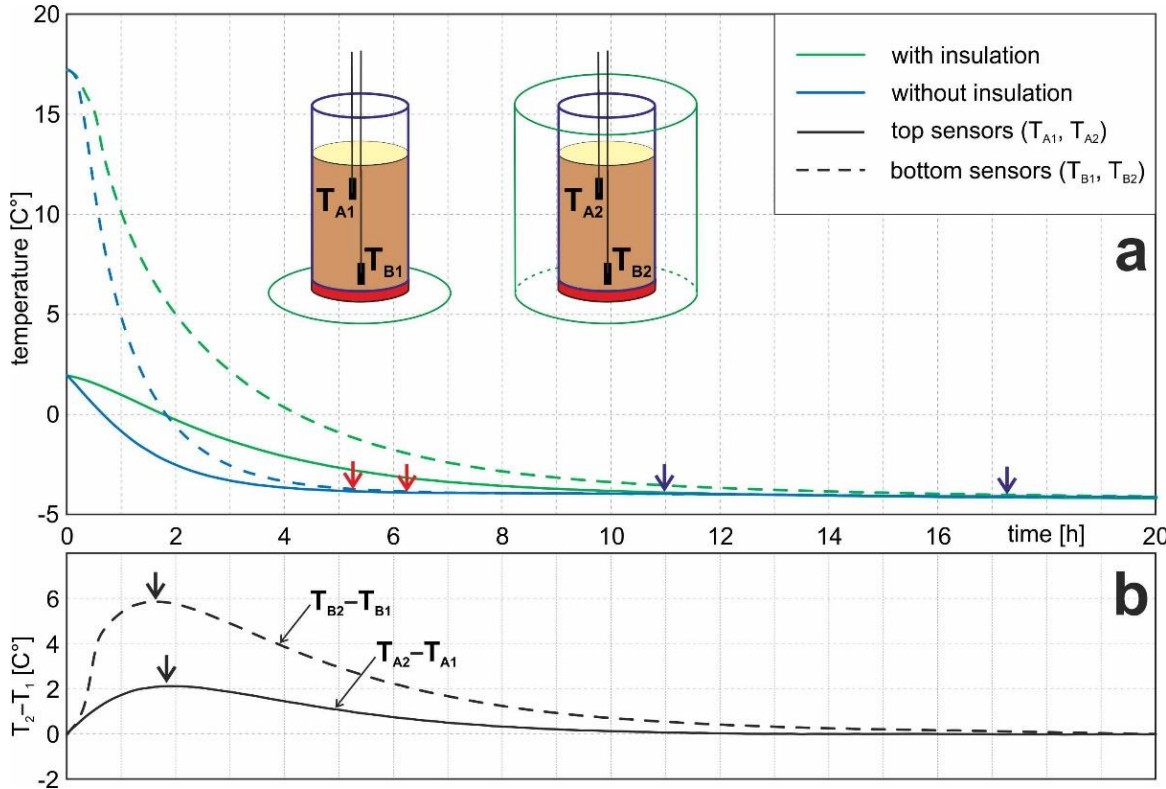

**Figure 6.** (**a**) Temperature versus time at two sensors posed at the top and bottom of the sample during the cooling phase with and without lateral thermal insulation of the casing. Arrows indicate the approximate timing for reaching the environment temperature. (**b**) Temperature difference between the two measurements versus time. Black arrows indicate the timing of the maximum differential temperature between the two sensors.

Independent from their initial thermal conditions and hence the differential temperature with the environment (at ca. $-4$ °C), both sensors in the experiments without the lateral thermal insulation clearly document a much larger rate of temperature drop, while ambient values are approached within few hours (red arrows in Figure 6). Moreover, notwithstanding the different position of the sensors, with $T_B$ being much deeper and more thermally protected within the sample, the maximum difference is similarly observed after 1–2 h (black arrows in Figure 6), therefore confirming the occurrence of comparable phenomena of lateral heat dispersion in the experiments lacking the lateral insulation. In contrast, when the sample case is laterally isolated in a proper way, heat dispersion occurs only vertically, and the temperature drop is much slower.

The efficiency of the lateral insulation was also analyzed by means of a thermal camera (model FLIR C5, visual camera 5 MP; thermal sensitivity < 70 mK), allowing to observe the heat distribution on the top surface of the casing and the lateral isolation material. An example of these tests is represented in Figure 7 clearly showing the quite homogeneous temperature on the top surface of the sample and the very high temperature gradient immediately outside the casing, where the temperature on the top surface of the insulation material sharply fades into the ambient temperature.

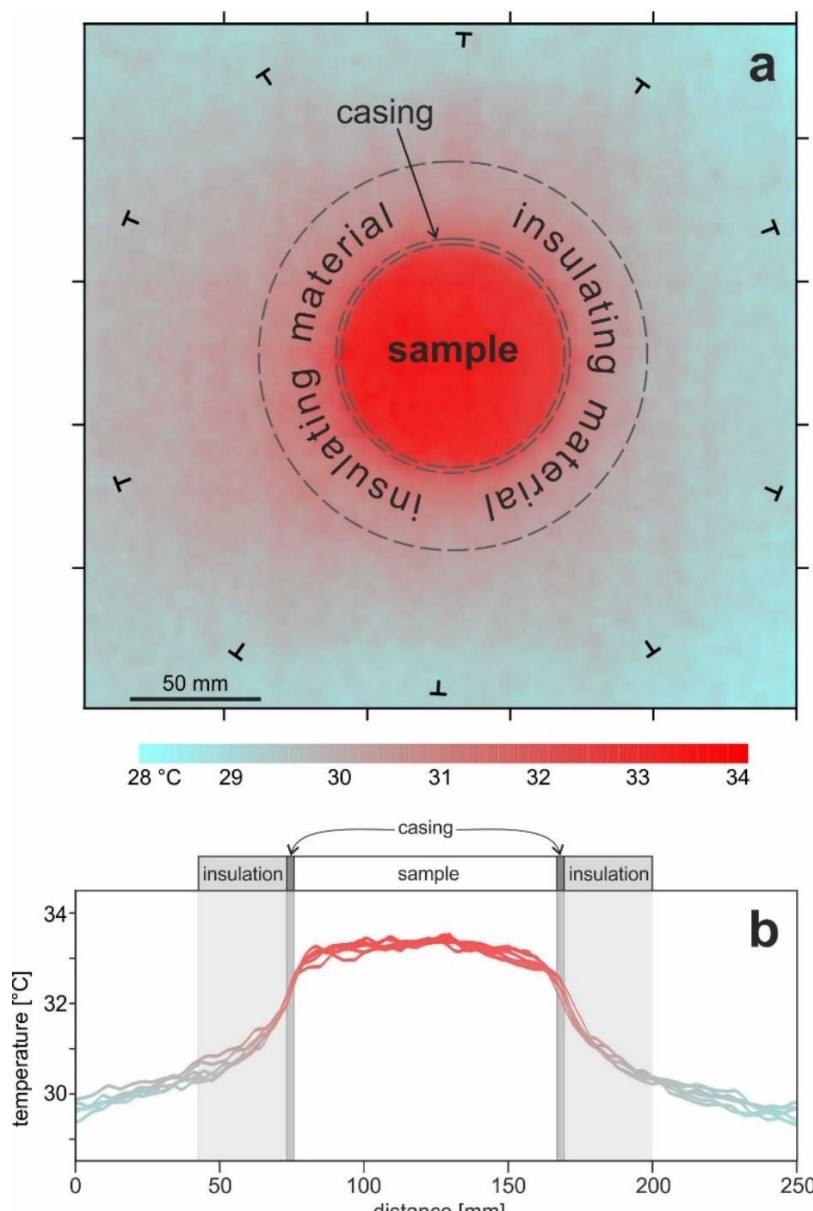

**Figure 7.** (**a**) Thermal image (top view) obtained during a test and some thermal profiles (**b**). The uniform temperature on the sample surface and the strong gradient outside the casing is evident, confirming the effective insulation and assuring the unidirectional heat flux during the experiments.

In conclusion, the performed tests satisfactorily confirm the efficiency of the designed and adopted thermal insulation strategy, both lateral and at the bottom of the casing, in order to assure a unidirectional heat flux across the samples during the experiments.

### 3.3. Reproducibility Test

A central issue in custom-built apparatuses is represented by the reproducibility of the results when several tests are performed with the very same material and thermal conditions. In order to verify this crucial aspect, we tested several times identical set-up conditions in terms of mineral composition, texture, porosity, water content, and heating power. The only difference observed during the diverse tests was the ambient temperature showing slight variations from test to test of 1–2 degrees (Figure 8).

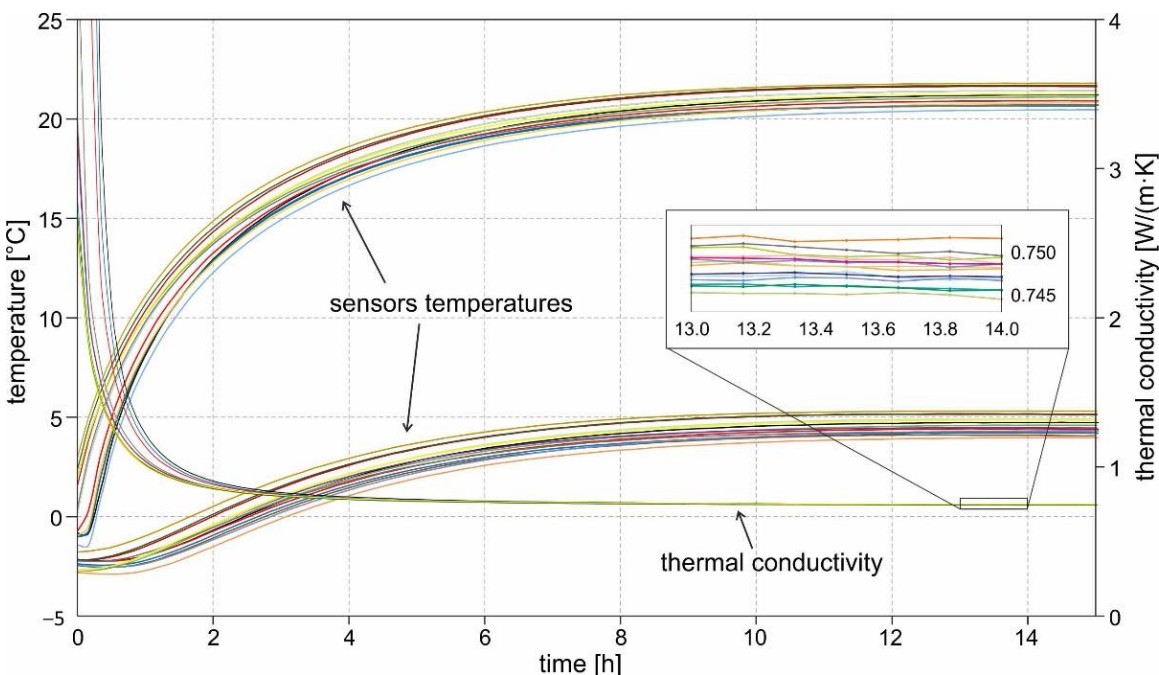

**Figure 8.** Measured temperatures (top and bottom sensors, left axis) and thermal conductivity (right axis) calculated from 15 experiments carried out for verifying the repeatability of the obtained values and hence the reliability of the built apparatus. See text for discussion.

Notwithstanding these operative variations during the different experiments, the thermal conductivity calculated by means of Equation (2) is spectacularly stable for all the tests (Figure 8). This is clearly evident from the curves represented in the inset of Figure 8. Indeed, the enlarged graphs of the calculated thermal conductivity show that during the steady-state regime, the obtained values of the investigated parameter could vary at the third decimal representative number ($\pm 0.002$ W/m·K), therefore documenting the repeatability of the measurements and the reliability of the created apparatus.

### 3.4. Comparison with Literature Data

Beyond the previously discussed tests, we also performed several other experiments for comparing the thermal conductivity values obtained from our apparatus with other published results of comparable materials. For this purpose, we selected from the literature laboratory experiments which have been carried out on sandy and silty-clayey sediments, in both dry and fully saturated conditions [1,11,17,18,24–26,39–43]. For the two granulometric classes, different authors commonly provide a range of values for the calculated thermal conductivity as represented in Figure 9a. The available information about mineralogical and textural characteristics are reported in the Supplementary Material. As can be observed, the results we obtained nicely fit the ranges of the λ values proposed by different authors. Slight differences or wider ranges by some authors are due to the different tested material either in terms of mineralogy and/or texture as well as sometimes on the different technique applied for drying the samples. For the aims of this paper, the results of the comparison are certainly satisfactory to confirm the reliability of the obtained thermal conductivity values.

We also performed several tests by varying the water content for sandy samples. In this case, we compared our results with some empirical relationships proposed in the literature correlating thermal conductivity to the degree of water saturation. Moreover, in this case, the results obtained with our apparatus with different percentage of water content are in perfect agreement with the proposed curves (Figure 9b).

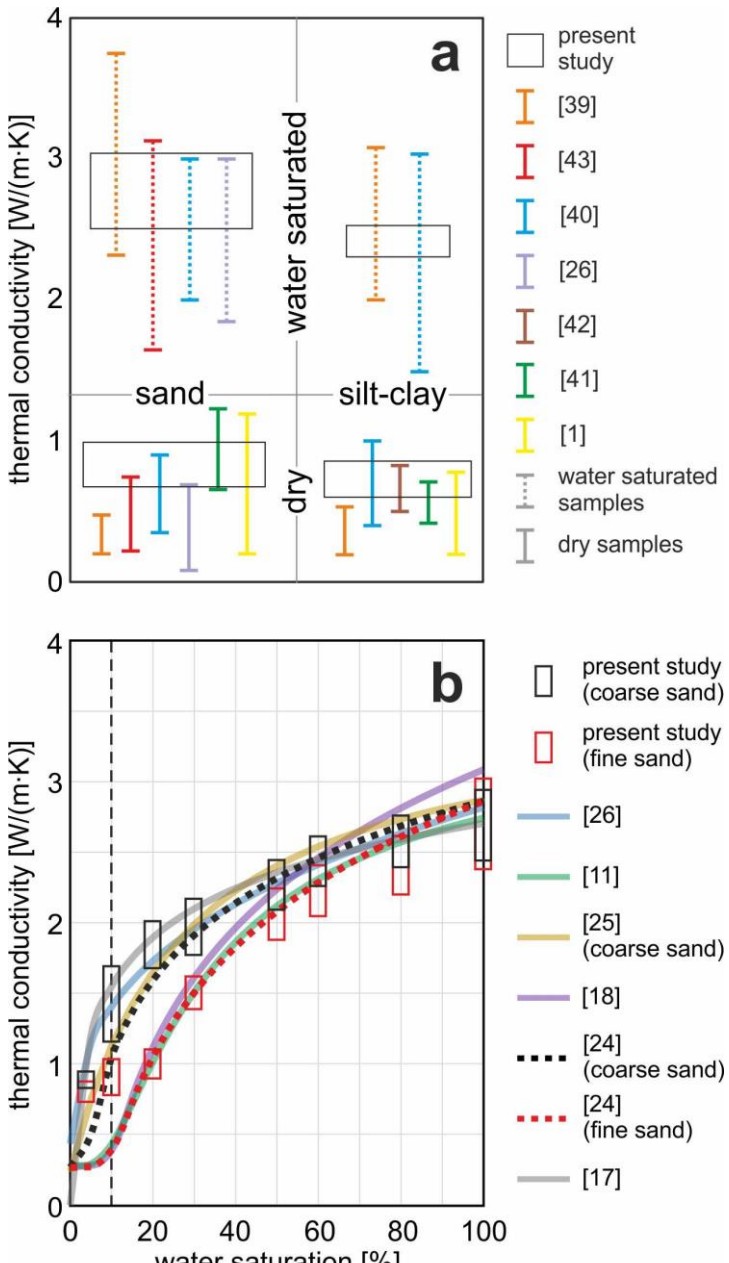

**Figure 9.** (**a**) Comparison of thermal conductivity values proposed in the literature for sandy (left) and silty-clayey (right) materials with some preliminary results obtained from our laboratory apparatus [1,26,39–43]. (**b**) Comparison between empirical relationships correlating thermal conductivity with water saturation in sands proposed in the literature and some of our preliminary tests on sands carried out with different water content [11,17,18,24–26]. The information provided by the different authors about the mineralogical and textural characteristics of the materials tested for thermal conductivity measurements are reported as Supplementary Material (Table S1).

## 4. Concluding Remarks

The present note is devoted to describing the set-up of a custom-built experimental laboratory apparatus for measuring the thermal conductivity of loose materials, both natural and artificial. The apparatus has been designed to operate under different conditions, e.g., ambient temperature, water saturation, and physico-chemical parameters of the fluids, as well as for testing different grain-size distribution and mineral composition of the grains. The custom-built laboratory is based on the guarded hot plate method, where a unidirectional thermal gradient is generated by a conduction process. A constant input power is

provided on one side of the sample as long as steady-state conditions are reached and a constant heat flux is established. Such a steady-state thermal regime is maintained for a sufficient time to obtain a stable value of the thermal conductivity (Figure 9). Accordingly, and based on fixed set-up parameters, e.g., the area crossed by the heat flux and the distance between the temperature sensors, the thermal conductivity is thus straightforwardly calculated by means of Equation (2).

The creation of an apparatus based on the guarded hot plate method, being the best technique for a scientific laboratory without taking advantage of existing commercial instrumentation, represented the major goal of the present research. In particular, the manually assembled hardware, including the complete remote monitoring and data logging, and the associated software make this instrument state-of-the-art and high-precision, as well as low-cost.

The strategy of each planning and building step was to create a device that could be exploited with different sample types, basically calibrated to test as many as possible different granular materials, both synthetic or natural ones, in a wide range of characteristics of the grains (granulometry distribution, porosity, mineralogical compositions, etc.) as well as being able to test different boundary conditions, such as environmental temperature or amount of water.

Assuring the easiness in preparing and managing the samples was another important target. For example, the amount of used material is ca. 1 dm$^3$ (less than 1 kg sand), and both the size and shape of the sample holder are comparable to most cores from the drilling of soils and shallow sedimentary successions.

As above mentioned, other experimental apparatuses are described in the literature for directly measuring thermophysical parameters from laboratory samples; only few of them can test sample dimensions comparable to ours [34,44], but most need larger volumes [33,35,36]. On the other hand, almost all of the proposed laboratory setups base their measurements on a radial heat distribution, thus applying the infinite line source method [33,34,36], while only [35] use a unidirectional heat flux (e.g., guarded hot plate approach) as we did. It should be noted, however, that the tested sample dimensions by the latter authors are far larger than ours. In summary, the most important differences between the instrumental setup presented in this paper and other ones probably consist in the overall low-cost of the instrumentation, its modularity with the possibility to test more than one sample at the time, its easy reproducibility by other research teams, and, contextually, the possibility of obtaining results comparable with previously published ones as documented in the previous sections.

Finally, we performed the swot analysis of the GeoTh experimental laboratory which is represented in Table 1.

**Table 1.** Swot analysis of the GeoTh experimental laboratory.

| Strengths | Weaknesses | Opportunities | Threats |
|---|---|---|---|
| • accurate measurements<br>• variety of material tested<br>• flexibility<br>• cheap<br>• remotely manageable | • introduction of human error in the preparation of the sample which could alter the natural conditions<br>• material wear and tear<br>• only conductive flux | • measurement/verification of the thermal conductivity of materials in dry and wet conditions<br>• optimal system exploration of the shallow geothermal systems | • increase in the costs of the construction materials of the system |

**Supplementary Materials:** The following supporting information can be downloaded at: https://www.mdpi.com/article/10.3390/soilsystems6040088/s1, Figure S1: A screenshot of the dashboard during a test. Table S1: Available information about the tested materials considered in this paper.

**Author Contributions:** Conceptualization of the project, D.R. and R.C.; methodology, D.R., A.M., I.N., M.A. and R.C.; laboratory data collection and data analyses, A.M.; hardware assemblage, I.N., M.A. and A.M.; software implementation, M.A. and I.N.; hardware and software validation, I.N., A.M. and M.A.; writing—original draft preparation, A.M., M.A. and I.N.; writing—review and editing, R.C and D.R.; funding acquisition, R.C.; general supervision, D.R. All authors have read and agreed to the published version of the manuscript.

**Funding:** The research activities of A.M. are supported by a grant of Italian MIUR in the frame of the Project "Dipartimenti di Eccellenza", while those of D.R. by a contract in the frame of the PON REACT EU Project by the Italian MUR.

**Institutional Review Board Statement:** Not applicable.

**Informed Consent Statement:** Not applicable.

**Data Availability Statement:** Intellectual protection policies of University of Ferrara and INFN do not allow to share at public domain level all details of the described system. On the contrary anyone could re-produce the system and make it commercial, using improperly work already done by Public Ital-ian Institution. People interested at the system or in a collaboration with this project, without profit interest, could contact authors and their institutions.

**Acknowledgments:** The authors wish to thank the technicians of the INFN laboratories at the Ferrara Section for their support.

**Conflicts of Interest:** The authors declare no conflict of interest as far as the assemblage of whole apparatus has been performed at the University and it has a purely scientific purpose.

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
