# Peer review of "GeoTh: An Experimental Laboratory Set-Up for the Measurement of the Thermal Conductivity of Granular Materials"

_soilsystems, doi:10.3390/soilsystems6040088_

Round 1
Reviewer 1 Report
The manuscript deals with a new low-cost experimental apparatus which can be addressed to measurements of the thermal conductivity of granular materials. The use of this apparatus could be very useful in several fields of research: soil science, agronomy, geoengineering and ground heat exchange, this latter producing thermal energy through borehole heat exchangers coupled with geothermal heat pumps. Nevertheless I realized a criticality in the manuscript which is the total lack of characterization of the sandy and silty-clayey materials used by the Authors for obtaining results of thermal conductivity from the proposed new apparatus. In the introduction the Authors say "the influence of the mineral composition, the bulk density and the porosity are the focus of an ongoing research project and the results will be published in a future paper"....Anyway, as the Authors make comparisons of thermal conductivity of the analysed granular materials with the new developped apparatus with those of sandy to silty-clayey samples from literature data (chapter 4.4) it should be explained, ALSO in the present paper:
- the type (geological origin, provenance) of the investigated granular materials, also comprising mineralogical composition, porosity and bulk density;
- general informations about mineralogical composition and physical properties (if available) of the sandy to silty-clayey material used for comparisons and shown in Figure 10, i.e. the materials used by Roshankhah et al (2020) Chen (2008), Lu et al (2007), Cote and Konrad (2005), Farouki, (1981), Johansen (1975), Kersten (1949), Barry Macaulay et al (2003), Nusier and Abu-Hamdeh (2000), Abu-Hamdeh et al (2001).
In addition, a short discussion concerning the comparison between the new proposed apparatus and methods of thermal conductivity measurements of granular material with comparable small size instruments already present from literature and cited in the introductory chapter (Kramer et al 2014, Giordano et al 2013, Dalla Santa et al 2017, Katsura et al 2006) will be very welcome.
Other comments:
- Figure 1 should be improved about the cylinder sample casing apparatus. Where is the aluminum film on the top in order to minimise water evaporation ? Please add in the figure the aluminum film. In addition in this figure also a graphic scale for the sample casing would be very welcome to have a glance on the diameter and height of the cylinder sample casing itself.
- Is figure 6 necessary? As it is a screenshot of the dashboard during a test (already explained in text) it could be better considering this figure as supplementary material?
Author Response
[in black the reviewer's comments; in blu our replies]
Reviewer 1
The manuscript deals with a new low-cost experimental apparatus which can be addressed to measurements of the thermal conductivity of granular materials. The use of this apparatus could be very useful in several fields of research: soil science, agronomy, geoengineering and ground heat exchange, this latter producing thermal energy through borehole heat exchangers coupled with geothermal heat pumps. Nevertheless, I realized a criticality in the manuscript which is the total lack of characterization of the sandy and silty-clayey materials used by the Authors for obtaining results of thermal conductivity from the proposed new apparatus. In the introduction the Authors say "the influence of the mineral composition, the bulk density and the porosity are the focus of an ongoing research project and the results will be published in a future paper".
As suggested by the reviewer, we properly modified and extended the text on this issue.
Anyway, as the Authors make comparisons of thermal conductivity of the analysed granular materials with the new developped apparatus with those of sandy to silty-clayey samples from literature data (chapter 3.4; ex 4.4) it should be explained, ALSO in the present paper:
- the type (geological origin, provenance) of the investigated granular materials, also comprising mineralogical composition, porosity and bulk density;
ok, we added as Supplemetary Material the key information about the analysed material for facilitating a comparison.
- general informations about mineralogical composition and physical properties (if available) of the sandy to silty-clayey material used for comparisons and shown in Figure 10, i.e. the materials used by Roshankhah et al (2020) Chen (2008), Lu et al(2007), Cote and Konrad (2005), Farouki, (1981), Johansen(1975), Kersten (1949), Barry Macaulay et al (2003), Nusier and Abu-Hamdeh (2000), Abu-Hamdeh et al (2001).
ok, we added the relevant information about the several materials that have been analysed by the different authors in the past, though with some difficulty and certainly some heterogeneity because in the corresponding papers not always a full information about mineralogical composition and physical properties is provided.
In addition, a short discussion concerning the comparison between the new proposed apparatus and methods of thermal conductivity measurements of granular material with comparable small size instruments already present from literature and cited in the introductory chapter (Kramer et al 2014, Giordano et al 2013, Dalla Santa et al 2017, Katsura et al 2006) will be very welcome.
ok, we added a comment and discussion relative to the comparison of our setup and other ones proposed in the literature.
Other comments:
- Figure 1 should be improved about the cylinder sample casing apparatus. Where is the aluminium film on the top in order to minimise water evaporation? Please add in the figure the aluminium film. In addition in this figure also a graphic scale for the sample casing would be very welcome to have a glance on the diameter and height of the cylinder sample casing itself.
done, we improved the figure as suggested by the reviewer.
- Is figure 6 necessary? As it is a screenshot of the dashboard during a test (already explained in text) it could be better considering this figure as supplementary material?
as suggested by the reviewer, we deleted figure 6 from the main text and moved it into the Supplementary Material (Figure S1).
Reviewer 2 Report
This article is an announce of the new experimental set up. It looks like an advertisement: "come to our labor and measure your sample with us". I do not think it is a good stile of work. I also do not think that declaration about the absence of conflict of interest is really true in this case.
The manuscript can become a real article and the conflict of interest issue will be closed if the installation is described in such a way that the reader can reproduce it himself and the software is attached as accompanying files.
There are some addition notes:
- Part 2. Materials and Methods does not contain a description of methods and materials. It is a part of the introduction (state of the art).
- Figure 1 has no title
- The images in figures 3, 4, 6, 8, 9 have insufficient resolution.
- Symbols in fig. 8 require reworking so that the reader can easily and unambiguously identify lines and shapes.
- What do the arrows in Fig. 7a?
- Figurative points on the fig. 10b may be divided by the groups: 1) points, corresponding Lu et al (2007) and Farouki (1981) lines 2) corresponding the Chen (2008) and Kersten (1949) lines. Why? Is it a result of two groups of samples?
The article does not provide a comparison of the results of thermal conductivity measurements for specific soil samples or sediments, for which it would have been previously measured by another previously generally accepted method (for example, using commercial systems). In fact, in the article the results of the changes are compared with model estimates, but with the quality of the match that we see in Fig. 8, the question arises: why measure, if calculations with such accuracy can simply be calculated from the models?
I believe that in order to verify the measurement quality, it is necessary to compare the results not with theoretical models, but with the results of measurements performed in a different, well-established way, ideally - in several ways, so that the conclusions about the measurement accuracy are substantiated.
Author Response
[in black the reviewer's comments; in blu our replies]
Reviewer 2
This article is an announce of the new experimental set up. It looks like an advertisement: "come to our labor and measure your sample with us". I do not think it is a good stile of work. I also do not think that declaration about the absence of conflict of interest is really true in this case.
The present publication is absolutely not intended to represent and advertisement, but we want to present and document the feasibility of building a low-cost experimental thermophysical laboratory that could provide correct results (in any case comparable with other ones published in the scientific literature), in a reasonable time, with a sufficient operating easiness. Accordingly, we firmly reject any suspect of conflict of interest.
The manuscript can become a real article and the conflict of interest issue will be closed if the installation is described in such a way that the reader can reproduce it himself and the software is attached as accompanying files.
The paper describes in details the measurement technique and the test sequence, then anyone could reproduce measurements and test using different hardware already present at his own laboratory. There is no need to attach the code prepared only for controlling the assembled hardware parts and recording the measured values. On the other hand, intellectual protection policies of University of Ferrara and INFN do not allow to share at public domain level all details of the described system. On the contrary anyone could reproduce the system and make it commercial, using improperly work already done by Public Italian Institution. People interested at the system or in a collaboration with this project, without profit interest, could contact authors and their institutions.
There are some addition notes:
- Part 2. Materials and Methods does not contain a description of methods and materials. It is a part of the introduction (state of the art).
correct, we modified the chaptering accordingly
- Figure 1 has no title
ok, we added the title as suggested
- The images in figures 3, 4, 6, 8, 9 have insufficient resolution.
done, all figures have been improved in terms of graphics and uploaded with a higher resolution.
- Symbols in fig. 8 require reworking so that the reader can easily and unambiguously identify lines and shapes.
ok, figure 7 (ex Fig. 8) has been improved as suggested by the reviewer
- What do the arrows in Fig. 7a?
as suggested we explained the meaning of the arrows in the caption
- Figurative points on the fig. 10b may be divided by the groups:1) points, corresponding Lu et al (2007) and Farouki (1981) lines2) corresponding the Chen (2008) and Kersten (1949) lines. Why? Is it a result of two groups of samples?
Represented curves are from empirical relationships as proposed in the literature on the basis of numerous experimental tests (not from simple numerical modelling), while circles are our experimental results; we think there is no real reason to subdivide the curves in two groups.
The article does not provide a comparison of the results of thermal conductivity measurements for specific soil samples or sediments, for which it would have been previously measured by another previously generally accepted method (for example, using commercial systems).
What does it mean “specific” soil samples? On the other hand, chapter 3.4 (ex 4.4) and Figure 9 (ex Fig. 10) do provide indeed a comparison of our experimental results of thermal conductivity with other results based on “previously generally accepted methods” as provided in the scientific (and commercial) literature.
In fact, in the article the results of the changes are compared with model estimates, but with the quality of the match that we see in Fig. 8, the question arises: why measure, if calculations with such accuracy can simply be calculated from the models?
Figure 7a (ex Fig. 8a) does not represent a “model” but it is a thermo-photograph of a real experimental test, where only colors have been modified for enhancing readability. Profiles in Figure 7b (ex Fig. 8b) are directly obtained from the digital version of the thermo-photograph and are not the results of numerical modelling but measured temperature values.
I believe that in order to verify the measurement quality, it is necessary to compare the results not with theoretical models, but with the results of measurements performed in a different, well-established way, ideally - in several ways, so that the conclusions about the measurement accuracy are substantiated.
In the article, we never refer to theoretical (or numerical) models and all our results as well as the ones considered for comparison are obtained from experimental measurements of real samples of natural materials.
Round 2
Reviewer 2 Report
Apparently, the authors took the words about the conflict of interest as an accusation. I did not mean to offend them at all, because a conflict of interest is not always a bad thing. For example, in this case, the authors of the text are both designers and owners of the installation they are talking about. It is this fact that I consider it necessary to indicate in the "conflict of interest" section.
But there is nothing discrediting the authors in this. I take my hat off to the people who were able to make measuring equipment out of simple and cheap components that work no worse than expensive ones, made by big corporations.
The authors have significantly improved the article compared to the first version. The article has become much better, but some refinement is steel needed. Some questions have appeared only now precisely because the article has become clearer.
1) The article lacks a formula by which the thermal conductivity was actually calculated. From general considerations, I do not see how this can be done without measuring the energy passing through the sample. And if this flow was measured, then how was it done. What device?
2) Figure 8 shows the dependence of thermal conductivity on time, decreasing from 4 to 0.745-0.75 in 12 hours. Meanwhile, thermal conductivity is a physical constant of matter. Obviously, at the initial moment of time, a more complex process takes place, when not only heat transfer occurs, but also energy is consumed to heat it (equations 1 and 2 are for the stationare conditions, they do not take into account the absorption of energy by matter (heat capacity). Then you need to call the right Y axis not "thermal conductivity", but "apparent thermal conductivity", or some similar term.
3) In Figure 8, the reproducibility of the results is 0.8%, while in Figure 9b, the difference between the results of measuring the thermal conductivity of water-saturated sand at the same water saturation differ by more than 30%. Why?
It is possible that different samples were measured. But what? There is neither a table with sample descriptions nor a table with results.
4) The authors provide a table with information about which samples were examined in the cited papers, but do not provide characteristics of the samples they measured.
5) Authors asked « What does it mean “specific” soil samples?» I mean that the instead the article contains information about reproducibility of the equipment (fig. 8), there is no information about trueness or at least interlaboratory reproducibility. In order to demonstrate this, it is necessary to give the results of measuring a sample, the thermal conductivity of which is known for sure or measured in another laboratory according to a generally accepted method. For example, take the same sand that was used in some work to which you refer.
6) Fugure 4 has have insufficient resolution: it is impossible to read some of the inscriptions. Maybe you need to increase the font of the inscriptions?
7) Figure 9 needs to be revised: it is very difficult to match the margins and lines in the figures with the legend. You probably need to use not only different colors, but also solid, dotted, dash-dotted lines and field borders. Make field borders more contrast: now in some cases it is not clear whether I see field borders or color transitions due to insufficient color quality of the image.
8) I think in figure 9 it makes sense to indicate the type of rock studied, as this was done for (Johansen, 1975). If the samples you examined are different breeds, then it makes sense to show them with different icons. Otherwise, it is not clear why there is a difference of 30 percent or more at the same water content.
Author Response
please, see attached file
Thanks and regards
